# Answering Unseen Questions With Smaller Language Models Using Rationale Generation and Dense Retrieval

## Abstract

When provided with sufficient explanatory context, smaller Language Models have been shown to exhibit strong reasoning ability on challenging short-answer question-answering tasks where the questions are unseen in training. We evaluate two methods for further improvement in this setting. Both methods focus on combining rationales generated by a larger Language Model with longer contexts created from a multi-hop dense retrieval system. The first method ($RR$) involves training a Rationale Ranking model to score both generated rationales and retrieved contexts with respect to relevance and truthfulness. We then use the scores to derive combined contexts from both knowledge sources using a number of combinatory strategies. For the second method ($RATD$) we utilise retrieval-augmented training datasets developed by Hartill et al. (2023) to train a smaller Reasoning model such that it becomes proficient at utilising relevant information from longer text sequences that may be only partially evidential and frequently contain many irrelevant sentences. We find that both methods significantly improve results. Our single best Reasoning model materially improves upon strong comparable prior baselines for unseen evaluation datasets (StrategyQA 58.9 → 61.7 acc., CommonsenseQA 63.6 → 72.7 acc., ARC-DA 31.6 → 52.1 F1, IIRC 25.5 → 27.3 F1) and a version utilising our prior knowledge of each type of question in selecting a context combination strategy does even better. Our proposed models also generally outperform direct prompts against much larger models (BLOOM 175B and StableVicuna 13B) in both few-shot chain-of-thought and standard few-shot settings.

## 1 Introduction

*"It was soon realized that the problem of systematically acquiring information from the environment was much less tractable than the mental activities the information was intended to serve" - Moravec (1988)*

Moravec's paradox is the observation that problems such as developing an ability to reason, that might have been assumed to be one of the most difficult challenges in artificial intelligence has been easier to resolve than the challenge of acquiring more basic knowledge such as sensory information. It is motivating to consider this in the context of recent advances in using both large Language Models (LLMs) and retrieval against large textual corpora for information acquisition in the question-answering domain.

We focus on methods to improve the performance of a smaller Language Model[1] (i.e. Reasoning Model) which, given a question and an acquired explanatory context as input, is expected to reason to provide an answer. Our interest in smaller models for this task is because we are interested in evaluating the viability of reasoning systems that answer arbitrary questions in resource constrained situations where available compute resource is limited, and internet connectivity and so forth is assumed to be unavailable.

To acquire the explanatory context, we consider two knowledge sources separately and in combination; retrieval of an explanatory context from a corpus of English Wikipedia paragraphs and rationale[2] generation

---

[1] Generative Transformers with 400 million to 1 billion parameters
[2] We use the term "rationale" to denote a free-text explanation (Wiegreffe & Marasović, 2021) of approximately one to three sentences that provides evidence to support a model prediction. We use the term to distinguish LLM generations of this form from the longer explanatory contexts produced from our retrieval system.

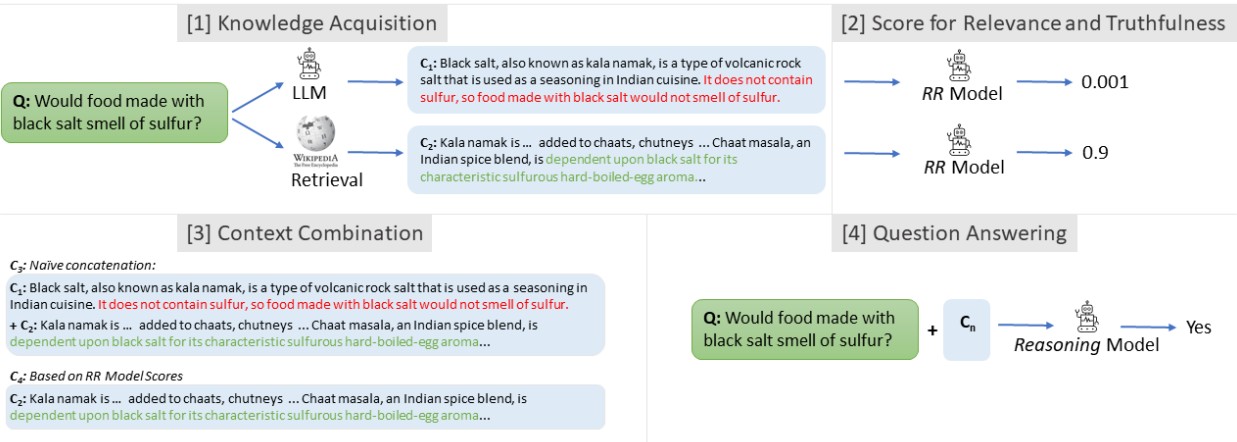

Figure 1: Overview of our approach. Given an unseen question **Q**: [1] we acquire explanatory contexts, **C₁** and **C₂**, from two knowledge sources. [2] We score the acquired contexts for relevance and truthfulness using a Rationale Ranking ($RR$) model that we train on diverse relevant/irrelevant samples that make both truthful and false assertions. [3] We evaluate and select methods for combining or filtering **C₁** and **C₂**. [4] We evaluate the performance of different contexts (**Cₙ**) on a set of Reasoning Models that are trained on different mixtures of training datasets, including a mixture containing $RATD$ datasets (Hartill et al., 2023) and a mixture without these. In the diagram, red denotes false information and green highlights relevant and truthful evidence.

from LLMs. Retrieval has generally been a relatively resource-efficient activity but until recently even inference on LLMs has required considerable computational resources. Recent innovations such as those involving 8-bit matrix multiplication (INT8) (Dettmers et al., 2022) enable the use of LLMs as frozen knowledge bases in constrained settings. For example inference on the 13 billion parameter StableVicuna model (Stability-AI, 2023) that we convert to INT8 and use in some experiments runs in approximately 18 GB of GPU RAM, well within the current capacity of large consumer GPU cards.

We choose retrieval from a reliable corpus and LLMs as our knowledge sources since we hypothesise that they may offer differing and complimentary characteristics. Studies such as Khattab et al. (2021); Hartill et al. (2023) have shown that multi-hop retrieval systems can be proficient at identifying the relevant $n$ documents necessary to answer $n$-hop factual questions where $n$ can be greater than two, e.g. those found in the Hover (Jiang et al., 2020) or Musique (Trivedi et al., 2022) datasets ("The Rhine forms a border between Aschenbrödel's composer's country and another country where women got the vote when?"). However we are unaware of any corresponding studies on LLMs that demonstrate similar proficiency in generating sufficient information to answer such $n$-hop questions. Conversely, it has been shown that LLMs can be strong at answering commonsense questions without using external retrieval (Lourie et al., 2021), but for such questions retrieval from large textual corpora offers limited benefit (Piktus et al., 2021; Hartill et al., 2023).

We explore two methods of combining information from our knowledge sources. Our Rationale Ranking method ($RR$) involves training a smaller Transformer to score both rationales and retrieved explanatory contexts with respect to relevance and truthfulness. We then evaluate a number of simple strategies to create combined contexts such as including either or both components that score over a threshold, or selecting the single top-scoring component. We focus on identifying combination methods that work best in the general case, i.e. are most likely to work well for an arbitrary unseen question for which we provide no means of predicting which combination method will work best. We find that we are able to identify such a method for each of our Reasoning Models and quantify the performance improvement (section 3.4.3). Our second method ($RATD$) consists of training our Reasoning Model with retrieval-augmented datasets previously developed by Hartill et al. (2023). These datasets were originally developed to impart diverse reasoning strategies such as an ability to identify and weigh partially evidential facts in long, noisy contexts. Noting

that where our rationales and retrieved contexts are combined, the resulting context is similar in length and form to the *RATD* contexts, we find that training on them enables a single Reasoning Model to utilise our various context formats effectively, including the case where the context consists of the naïve concatenation of rationale and retrieved context that does not consider the *RR* model scores.

In summary the major contributions of this paper are: (A) We propose *RR*, a novel method that both selects context components by relevance, and filters components that may be false. (B) We apply the *RATD* method developed by Hartill et al. (2023) to facilitate reasoning over contexts that potentially combine information from multiple knowledge sources. (C) We demonstrate that both methods in isolation significantly improve reasoning performance in smaller Language Models from strong baselines in the same unseen setting (section 3.4.3). (D) We show that smaller Language Models can manifest comparable or stronger reasoning performance as a LLM when provided with the same knowledge to reason over that the LLM is capable of generating for itself (section 3.4.2). (E) We illustrate the respective strengths and weaknesses of LLMs and multi-hop retrieval from a Wikipedia corpus as knowledge sources (section 3.4.2). (F) We show that combining information from these sources significantly improves the overall average performance versus using a single source, often beyond what either knowledge source in isolation can deliver on individual datasets (section 3.4.2).

## 1.1 Related Work

**Knowledge Augmentation from LLMs**. Bosselut et al. (2019) proposed COMET, a GPT-based Model (Radford et al., 2018) trained on triples from the ATOMIC (Sap et al., 2019) and ConceptNet (Speer et al., 2017) knowledge graphs such that it would generate potentially novel triple completions. Shwartz et al. (2020) compare augmentation methods using COMET, ConceptNet and their self-talk method where the question-answering Language Model is self-queried to produce additional information pertinent to answering the question. Liu et al. (2022) generate knowledge statements from GPT-3 (Brown et al., 2020) conditioned on the question and use the augmented samples in separate smaller Reasoning Models. Following the introduction of chain-of-thought (COT) prompting (Wei et al., 2022), a number of recent papers use this prompting style to distill training sets of rationale-augmented samples from internet-accessible LLMs (GPT-3, Palm (Chowdhery et al., 2022)) which are then typically used to train much smaller models in task-specific finetuned settings e.g. (Magister et al., 2023; Li et al., 2023; Hsieh et al., 2023; Wu et al., 2023; Shridhar et al., 2023) sometimes such that the label and the rationale are output to avoid the issue of having to generate a rationale from the LLM at test time. We note that our usage of LLM-generated rationales is rather different from these in that we assume a locally-accessible LLM (with lower resource requirements) at test time and do not incorporate LLM-generated rationales in our Reasoning Model training.

**Retrieval from Textual Corpora**. For a comprehensive introduction to this wide field we suggest reviewing Lin et al. (2022a) and (Mitra & Craswell, 2018). In summary, TF-IDF (Spärck Jones, 1972) has been used for many years to associate queries with documents using adjusted bag-of-word count vectors. This approach carries the advantage that fine-tuning for the target dataset is not required. Chen et al. (2017) first used such sparse retrieval against Wikipedia in the context of open domain question-answering. In dense retrieval, query and corpus documents are embedded into the same vector space with similarity defined as the inner product between a query and a document vector. Karpukhin et al. (2020) used dense retrieval to identify a single document sufficient for answering a single-hop question. Izacard et al. (2022) reduce the need for target dataset finetuning by pretraining a dense retriever on self-supervised data. Xiong et al. (2021) extend the dense retrieval approach to to retrieve two documents necessary to answer a complex two-hop question. Hartill et al. (2023) extend this to enable retrieval of an arbitrary maximum number of documents (in practice $n \leq 4$). Wang et al. (2018) introduced a Reranker Model that refines retrieved results. Baleen (Khattab et al., 2021) is a two-stage condenser system comprising a Reranker followed by an additional model that scores relevance of each sentence selected over multiple documents ($n \leq 4$) from the first stage. Hartill et al. (2023) introduce an Evidence Set Score into the second stage to quantify the sufficiency of the entire set of selected sentences for answering a query and call their resulting system the "Iterator". As noted, in this paper we use the Iterator with a Wikipedia corpus as described the following section.

**Multiple Knowledge Sources**. Retrieval has been successfully used as a method for querying knowledge graphs by embedding the constituent triples as the document vectors in addition to, or instead of, standard

text, e.g. Yu et al. (2022) augment commonsense questions with retrieved information from a commonsense-focused corpus consisting of information source from knowledge graphs, commonsense datasets and other textual sources. Perhaps most similar in spirit to our work Pan et al. (2023) consider knowledge graphs, Wikipedia data, a dictionary, and others, as separate knowledge sources, each queried using dense retrieval. In contrast to our approach of considering various methods for combining information, they train a model to select the single most relevant source for augmenting each input sample. This is analogous to our "Max Score" method described in section 3.3. Like us they train a smaller Reasoning Model with disparate training and evaluation datasets, although unfortunately their evaluation datasets differ from ours. Also in a similar direction to our "Max Score" method, Si et al. (2023) route a query to four expert LLMs and select the single most likely answer using a smaller classifier trained for that purpose. In a finetuned setting, Xu et al. (2022) also consider multiple knowledge sources. Here they use an entity linking method to query ConceptNet and sparse retrieval over a dictionary and a set of commonsense datasets. The results are always concatenated which is similar to our Naïve Concatenation method (section 3.3).

**Falsehood Detection**. Our $RR$ Model, trained to score for truthfulness and relevance over instances from disparate knowledge sources, can be seen as a novel extension to a Reranking approach. However it also shares an objective with methods aiming to detect falsehood in LLM generations. Generally these methods fall into three categories. The first are methods based on the intuition that higher token log probabilities correspond to better text along a particular dimension such as truthfulness (Yuan et al., 2021; Fu et al., 2023). The second are factuality detection methods that evaluate LLM-generated assertions as true if they can be supported by a external reference (e.g fact retrieval from a reliable corpus). Recent studies here include (Min et al., 2023; Chern et al., 2023). A third category, broadly called self-checking involves prompting a LLM such as ChatGPT or GPT-4 (OpenAI, 2023) to recognize their own errors (Chern et al., 2023), or refine their own outputs (Chen et al., 2023; Madaan et al., 2023), without recourse to external tools. In this category but with a different approach, Manakul et al. (2023) score the consistency between a reference statement and several stochastically sampled versions of it that may be likely to diverge more if the reference is a hallucination.

## 2 Method

An overview of our approach is provided in Figure 1. In following sections we describe how the two knowledge sources are implemented, how the $RR$ model is constructed, trained and initially evaluated, and how the Reasoning Models are trained. We describe our context combination methods further below in section 3.3 so as to make clear the nomenclature we use in reporting experimental results.

A major assumption is that our system may be asked arbitrary questions from unknown distributions. Therefore we primarily consider our evaluations in the unseen rather than fine-tuned setting. The most relevant comparisons we have available to us are the baselines for StrategyQA (Geva et al., 2021), CommonsenseQA (Talmor et al., 2019), ARC-DA (Bhakthavatsalam et al., 2021), IIRC (Ferguson et al., 2020) and Musique (Trivedi et al., 2022) established for smaller Language Models in unseen settings by Hartill et al. (2023). The datasets cover a diversity of question types requiring diverse reasoning strategies to answer, including commonsense and $n$-hop factual questions ($n \leq 4$) as discussed further in section 3.2. Hence we adopt these datasets for evaluation and use the same definition as Hartill et al. (2023) for "seen-ness" whereby an unseen evaluation sample is one from a dataset that is disjoint from any training dataset. In our case we extend this to our LLM generations, ensuring that all examples in few-shot prompts come from our training rather than evaluation datasets, or are manually created by us.

Aside from the baseline results, Hartill et al. (2023) also provide their "Iterator" $n$-hop dense retrieval system (where $n \leq 4$). In a single-hop retrieval model, samples are processed as (1) Input $\langle q \rangle$ with an objective of minimizing distance to the vector representing $d_0$ (hereafter denoted $\langle q \rangle \rightarrow d_0$, where $q$ and $d_t$ are the input question and the $t\text{-}th$ supporting document of $q$ to retrieve respectively). For a two hop system, the second hop is then (2) $\langle q, d_0 \rangle \rightarrow d_1$. In the Iterator model this is extended up to 4 hops i.e. $\langle q, d_0, d_1, d_2 \rangle \rightarrow d_3$.

We adopt this system as our "retrieval" knowledge source and re-use the retrieved contexts that are provided, both for $RATD$ datasets and for each evaluation dataset (section 2.2). Hartill et al. (2023) also provide a Reasoning Model that is trained in a multitask manner on a large number of datasets including their $RATD$

datasets. We train two additional Reasoning models in the same manner as Hartill et al. (2023) with, and without, the *RATD* datasets (section 2.4). By reusing all of the above components we are able to quantify the effect of adding the second knowledge source under both the *RR* and *RATD* methods versus the baselines established by Hartill et al. (2023) (section 3).

### 2.1 Rationale Generation

We utilize two LLMs, BLOOM BigScience Workshop et al. (2022) and StableVicuna (Stability-AI, 2023), a much smaller model that has been further tuned from the Vicuna v0 13B model (Chiang et al., 2023) which in turn was adapted from the LLama Touvron et al. (2023) foundation model. We chose these two models because they are representative of differing approaches to developing LLMs and they may offer divergent characteristics in rationale generation. At 176 billion parameters, BLOOM is the largest language model we had access to at the time that we could run under INT8. It is trained on 410 billion tokens and the version we used did not undergo further training on instructional data or human feedback. Llama by contrast is trained on one trillion tokens. From the Llama checkpoint, Vicuna underwent further training on user-provided ChatGPT conversations. Finally StableVicuna was developed from Vicuna by further training in both supervised and reinforcement learning from human feedback (RLHF) Ouyang et al. (2022) settings on a mixture of the human-generated OpenAssistant Conversations Dataset Köpf et al. (2023) and human-LLM conversations from the GPT4All Anand et al. (2023) and Alpaca Taori et al. (2023) projects. We used StableVicuna under both INT8 and FP16 versions, the former offering a smaller GPU memory footprint at around 18GB while the latter uses almost twice as much memory but we find inference much faster, thus offering a clear trade-off in a resource-constrained setting.

To generate rationales from each model, we used greedy decoding on chain-of-thought (COT) prompts (Wei et al., 2022) to generate the rationale followed by the phrase "So the answer is" and the answer (examples are in appendix B). This enabled us to evaluate the LLM answers directly from the same prompts and with the same rationale that our Reasoning Model would use, allowing a comparison under a similar set of assumptions. Occasionally a model would fail to generate the separate answer. In this case, to be favorable to the direct LLM method, the full rationale was used as the answer in calculating metrics. Generated rationale length is a maximum of 128 tokens.

To maintain the integrity of our unseen settings we ensured that no examples used in prompts were from any of our evaluation datasets. The prompts used were identical between our LLMs excepting that examples for StableVicuna prompts are denoted as:

`### Human: [question] ### Assistant: [rationale]. So the answer is [answer].`

BLOOM prompts are denoted as:

`Q: [question] A: [rationale]. So the answer is [answer].`

Our qualitative examination of rationales generated by BLOOM and StableVicuna suggests a diversity in quality from both models but that they tend to produce better rationales on the same datasets (e.g. ARC-DA) and worse on the same (e.g. Musique). We observed that BLOOM was generally more prone to generating falsehoods. Examples from both models may be found in appendix C. We note that robust examination of rationale quality is presently challenging to perform and believe research into automated methods in this area represents a promising future direction.

### 2.2 Retrieval

As well as the $n$-hop retrieval model discussed above, the Iterator also comprises a two-stage reranking system. The first stage is an $n$-hop Paragraph Reranker that scores retrieved paragraphs and sentences within paragraphs for relevance to the query at the current hop e.g. input $\langle q, d_0, d_1 \rangle$ to score $d_1$ on hop 2. Top-scoring sentences are passed to a second stage Evidence Set Scoring model that re-scores each sentence in the context of the accumulated set of top-scored sentences to the current hop (Evidence Set) as well as scoring the overall relevance of the Evidence Set.

For our "retrieval" knowledge source, as noted we simply reuse contexts generated by the Iterator, both for each evaluation sample and also for the creation of $RATD$ datasets for the training regimes. Iterator-generated contexts are formatted as a list of paragraph fragments that are recovered from the top-scored sentences, each prepended by the title of the corresponding document and containing the top-scoring sentences along with preceding and successor sentences where these exist. The top-scored sentences are identified by taking the Evidence Set from the top-scored hop. Contexts contain as many fragments as will fit into a 512-token sequence length. They are semi-structured as follows:

```
[Doc 1 title]: [One to three sentences from document 1 paragraph]. [Doc 2 title]: ...
```

The corpus utilised by the Iterator is obtained from the August 1 2020 English Wikipedia dump and consists of approximately 35 million paragraphs.

### 2.3 Rationale Ranker

Our $RR$ model takes a question and context pair as input $\langle q, c \rangle$ and produces a score $s$. It is trained with a binary cross-entropy objective where samples are labelled 1.0 if $c$ is truthful and fully evidential in answering $q$ or 0.0 otherwise. The model is trained on a mixture of existing datasets for which we acquire or construct positive $c$ (i.e. a set of relevant and truthful gold sentences that are sufficient to answer $q$), and negative $c$ (which omit some or all gold sentences and may be irrelevant, false or both with respect to $q$ answerability). We used shared normalization (Clark & Gardner, 2018) such that each $q$ is sampled in the same batch paired with a positive and separately a negative $c$. We found that without shared normalization, model training would collapse and it would predict every $c$ as negative. This may have occurred because without seeing positive and negative $c$ for the same $q$ in the same batch the pattern to be learned is insufficiently signalled.

Table 1: $RR$ model training dataset composition. The construction methods denoted "... facts" involve creating rationales from gold sentences or structured triples sourced from the cited study. Iterator-like contexts and Rationale-like are constructed from the training datasets' gold (and associated negative) paragraphs. LLM-sampled and LLM-greedy contexts are negative rationales generated by BLOOM using nucleus sampling and greedy decoding respectively. [a]Onoe et al. (2021); [b]Yang et al. (2018); [c]Thorne et al. (2018); [d]Khot et al. (2020); [e]Clark et al. (2016; 2018); [f]Jiang et al. (2020); [g]Inoue et al. (2020); [h]DeYoung et al. (2020); [i]Jhamtani & Clark (2020); [j]Xie et al. (2020)

| Training Mixture | Positive Contexts | | Negative Contexts | |
| --- | --- | --- | --- | --- |
| | Count | Construction Methods | Count | Construction Methods |
| Creak[a] (Commonsense) | 10173 | Creak facts[a] | 81408 | LLM-sampled |
| HotpotQA[b] (Multi-hop factual) | 34304 | R4C facts[g], Iterator-like, Rationale-like | 41839 | LLM-sampled, LLM-greedy, Iterator-like, Rationale-like |
| FEVER[c] (Single-hop factual) | 60986 | Eraser facts[h], Iterator-like, Rationale-like | 121427 | LLM-sampled, Iterator-like, Rationale-like |
| QASC[d] (Multi-choice science) | 47830 | QASC facts[d], eQASC facts[i] | 193214 | LLM-sampled, LLM-greedy |
| ARC[e] (Multi-choice science) | 6469 | WorldTree facts[j] | 24492 | LLM-sampled, LLM-greedy |
| Hover[f] (Multi-hop factual) | 28171 | Iterator-like, Rationale-like | 28171 | Iterator-like, Rationale-like |
| **Total** | **187933** | | **490551** | |

Since the model must score both rationale-style $c$ and Iterator-generated $c$ on the same scale, we develop training samples that are similar to both types. Obtaining positive $c$ for training questions is generally straightforward. These are constructed from gold sentences and paragraphs associated with each dataset. Negative $c$ that cover both irrelevance and falsehood are harder to obtain. We construct negative $c$ by two methods; (1) generating them from BLOOM using specially constructed few-shot prompts containing examples of negative rationales (e.g. appendix D), and (2) creating them synthetically by substituting gold sentences with negative ones using datasets such as HotpotQA that come with sentence level annotations. The synthetic method can only produce irrelevant negatives whereas the LLM-generating method produces both irrelevant and false rationales. For LLM generation we use both greedy decoding and nucleus sampling (Holtzman et al., 2019) to create negatives. We find that greedy decoding produces positive-appearing but negative samples but (obtusely) the LLM has a tendency to produce accidentally positive rationales which

we must filter out[3]. Nucleus sampling by contrast (temperature=0.95 and p=0.96) produces a diversity of false and irrelevant samples that are less likely to be accidental positives. However here falsehoods tend to have an exaggerated quality which could make them less adversarial for the model, so we create samples via both decoding methods (examples in appendix E). Dataset construction is summarised in Table 1.

We employ diverse combination methods involving the trained $RR$ model scores to create contexts for our evaluation datasets that combine rationales and Iterator-generated contexts, as described in section 3.3.

### 2.3.1 Rationale Ranker Evaluation

Our $RR$ development set consists of 89,470 samples taken from the respective development splits of our training datasets. Contexts are created using the same methods as illustrated in Table 1 for corresponding training splits. We sample a single positive or negative context for each development question such that there are equal positive and negative contexts. As shown in Table 2, accuracy is high in this in-domain setting.

Table 2: $RR$ model Accuracy on the in-domain development set (score threshold $t = 0.5$). Total is micro-accuracy. High accuracy is attainable in detecting both positive and negative contexts.

| Positive Context | Negative Context | Total |
|---|---|---|
| 91.5 | 93.0 | 92.3 |

Table 3: Accuracy in detecting falsehoods on TruthfulQA MC1. The $RR$ model is better at detecting falsehoods than the Iterator Paragraph Reranker which was trained to detect relevance but not falsehood. It's performance is competitive or better than much larger models that have not been trained using RLHF [a]OpenAI (2023); [b]from Lin et al. (2022b) Github repository; [c]model from Hartill et al. (2023) with results calculated by us.

| Model | TruthfulQA MC1 |
|---|---|
| GPT-4 RLHF[a] | 60.0 |
| GPT-3.5 RLHF[a] | 47.0 |
| GPT-4 No RLHF[a] | 30.0 |
| GPT-3 175B[b] | 21.0 |
| GPT-J 6B[b] | 20.0 |
| UnifiedQA 3B[b] | 19.0 |
| Iterator Paragraph Reranker 335M[c] | 18.2 |
| Rationale Ranker 335M (Ours) | 30.0 |

Turning to an unseen setting, we initially evaluate context relevance scoring with a five-way multi-choice relevance detection dataset that we create from the gold rationales supplied with StrategyQA (SQA), where the four incorrect options are simply randomly assigned rationales from other SQA questions (we use SQA since this is not part of $RR$ model training). Here our model achieves 91.4% accuracy. A more interesting question is the extent to which our relatively small $RR$ model is capable of detecting falsehoods in an unseen setting. To evaluate this question we consider TruthfulQA (Lin et al., 2022b), an adversarial evaluation-only dataset of 817 questions that models and/or humans tend to answer falsely. In Table 3 we compare falsehood detection performance of the $RR$ model with various larger models and in particular with the Iterator Paragraph Reranker. We treat the Paragraph Reranker as representative of models specifically trained to score context relevance but that have not necessarily been trained to consider truthfulness. We utilise the TruthfulQA MC1 split which is formatted as 4-5 way multi-choice with one truthful option. Each option is scored independently of other options and the highest-scoring selected as the prediction. In the

---

[3]We eliminate rationales where the stemmed text contains the stemmed answer string, excepting samples with yes/no labels. We use the snowball stemmer from NLTK (Bird et al., 2009).

case of LLMs the score is calculated as the log-probability of the completion following the question. For the Paragraph Reranker and our $RR$ model we use the score that each model has been trained to compute. It can be seen that the $RR$ model is indeed much better at detecting falsehoods than the Paragraph Reranker and it's performance is competitive or better than much larger models that have not been trained using RLHF. We imagine the superior performance of LLMs trained with RLHF on falsehood detection is due to their associated large reward models, like our $RR$ model, being trained in part to rate samples making false assertions as undesirable.

## 2.4 Reasoning Models

We consider three Reasoning Models in our experiments. Reasoning Models take a question and context pair as input $\langle q, c \rangle$ and generate an answer $a$. The first, which we use as a baseline, is the unmodified *"Base+RATD"* model from Hartill et al. (2023) which we denote here as the $RATD$ model for brevity. This is a multitask-trained model which is further trained from the original BART (Lewis et al., 2020) pretrained checkpoint on a large number of datasets[4]. For descriptive purposes, we divide these training datasets into two sets. The first are the $RATD$ datasets described in section 2.2, whose purpose is to confer an ability to reason over long, noisy, and partially evidential contexts. We denote the remaining large number of training datasets as the *Common* set; these broadly cover tasks designed to instill simple numerical literacy, and diverse question-answering ability. Hence we say that the $RATD$ model is trained on $Common \cup RATD$ datasets.

We create an additional set of training samples denoted $GR$ (for "gold rationales"). These are intended to impart further ability to reason over rationale-form contexts. $GR$ consists of samples for Creak, QASC, ARC, HotpotQA, and FEVER where the contexts are gold rationales constructed similarly and from the same sources as those described for the $RR$ model training dataset in Table 1.

We then develop our two main Reasoning Models, both multitask-trained using the same approach and hyperparameters as the original $RATD$ model: The $GR$ model is trained on $Common \cup GR$, and the $GR+RATD$ model is trained on $Common \cup GR \cup RATD$.

# 3 Experiments

## 3.1 Models

The Rationale Ranker is built upon ELECTRA-large (Clark et al., 2020). Reasoning Models are based on BART (Lewis et al., 2020). All models use the the Huggingface (Wolf et al., 2020) implementations. The Reasoning Models differ only in their respective training data; hyperparameters are otherwise identical.

## 3.2 Unseen Evaluation Datasets

All evaluation dataset results are reported against the same splits used by Hartill et al. (2023). As with that paper we use the numeracy-focused F1 calculation introduced in Dua et al. (2019) for ARC-DA, IIRC and Musique.

**StrategyQA** (Geva et al., 2021) (SQA) contains commonsense samples involving diverse multi-hop reasoning strategies with yes/no answers (average $n = 2.33$). The full training set is used for evaluation as with BIG-bench (Srivastava et al., 2022).

**CommonsenseQA** (Talmor et al., 2019) (CSQA) is a multi-choice dataset of commonsense questions derived from Conceptnet (Speer et al., 2017). The task is to choose the best option from five options of which more than one may sometimes be plausible.

**IIRC** (Ferguson et al., 2020) contains factual questions and an initial explanatory paragraph for each which must be augmented with additional retrieved information to be fully evidential ($1 \leq n \leq 4+$). Answers may be numbers, binary, text spans or labeled unanswerable.

---

[4]We refer the reader to Hartill et al. (2023) for a more exhaustive description of the training regime and dataset construction.

**ARC-DA** (Bhakthavatsalam et al., 2021) is a subset of ARC (Clark et al., 2018) (science questions) where questions have been re-worded to make sense in an open domain context. The original multichoice versions of ARC are part of our training regime for both Reasoning and $RR$ models, so samples are "partially unseen" in the sense that the question format is different.

**Musique** (Trivedi et al., 2022) is a $n$-hop factual dataset ($n \leq 4$) constructed by combining single-hop questions from existing datasets. The training split of Musique is used in all of our Reasoning Models, and in the Iterator training. However as with Hartill et al. (2023), we use the original development split as "partially seen" since development samples were constructed such that no single hop question, answer, or associated paragraph is common to the corresponding element of any training sample. Hence the form of questions is "seen" but the exact questions are not.

### 3.3 Context Combination Methods and Experimental Nomenclature

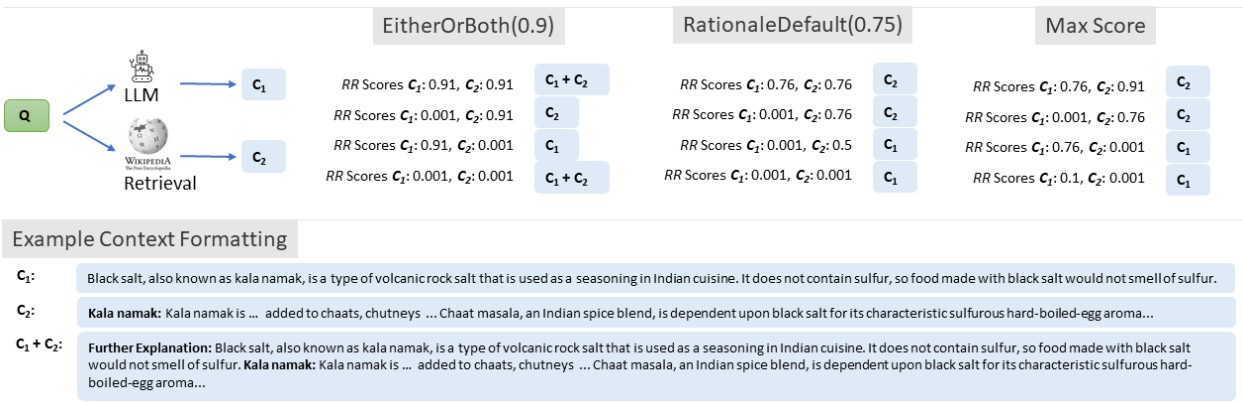

Figure 2: Examples of combining contexts. For a question **Q**, we acquire two contexts, **C₁** and **C₂**. The resulting combined context for our combination methods with example thresholds and $RR$ model scores is then shown in blue boxes where "+" denotes the concatenation of **C₁** and **C₂**. The Naïve Concatenation is always **C₁ + C₂**. Formatted examples of resulting contexts are shown at the bottom of the figure with titles shown in bold for readability. The phrase "Further Explanation" is added to the rationale in a concatenated context to mimic a document title.

For each unseen evaluation question, given a LLM-generated rationale, and an Iterator-generated context as possible combined context components, and $RR$ model scores for each, we evaluate methods of combining components. We implement four combination methods and create versions of our unseen evaluation datasets with combined contexts for each as follows:

**Naïve Concatenation**: The simple concatenation of a rationale and corresponding Iterator-generated context with the above form. $RR$ model scores are ignored.

**Max Score**: Choosing the single component that the $RR$ model scores highest.

**RationaleDefault**: Defaulting to taking the rationale component unless the Iterator component scores over a threshold $t$ in which case it is exclusively selected.

**EitherOrBoth**: Selecting either or both components that score over a threshold $t$. If neither component is selected, we default to the Naïve Concatenation context since smaller Language Models have been shown to be ineffective for answering unmemorized question-only (open domain) questions (Lewis et al., 2021).

For the latter two combination methods we create contexts using each of eight $RR$ score thresholds ranging from $t = 0.0005$ to $t = 0.9$. We denote the particular version using the threshold e.g. EitherOrBoth(0.9) means samples are augmented using the EitherOrBoth method with $t = 0.9$. Obviously innumerably other combination methods are possible but we find that this set is sufficient for our research purposes while remaining manageable. Figure 2 illustrates examples of contexts derived from each combination method

using hypothetical *RR* scores. Combined contexts are truncated (from the Iterator component) to the maximum sequence length of the model (512 tokens) at inference time.

Each of our three Reasoning Models might be expected to perform better with particular context types. For example the *GR* model might do better where the context tends to be rationale-like whereas the *RATD* model may do better where the context is of Iterator-generated form. This influences which combination method is likely to perform better on each Reasoning Model.

Similarly, different combination methods are likely to work better for differing question types (commonsense, multi-hop factual etc). For example knowing that LLM-generated rationales tend to be more effective than Iterator-generated contexts for answering commonsense questions, we can deduce that RationaleDefault(0.9) is likely to be a good strategy for developing contexts for CommonsenseQA because using this strategy results in Rationale-only contexts except where the Iterator context is scored very highly. However, we are interested in the situation where our model is presented with an arbitrary question of unknown type. Hence we are more interested in finding combination methods that will *generally* work well under this assumption, even where the method may not be the best for any particular type. We identify combination methods satisfying this criteria as those with the highest *unweighted macro-average score over our unseen evaluation datasets* (henceforth "Mean" or "Mean score") on each Reasoning Model, taking inspiration for averaging over heterogeneous metrics from e.g. Wang et al. (2019b;a). For the methods that utilize *RR* model scores we select the highest performing on this measure and refer to it as "Generally best RR combo" below. We also report the "Best RR combo per dataset" where we select the highest scoring combination method for each evaluation dataset. We note that since we cannot use this approach on an arbitrary question of unknown type we don't consider it a usable method in a truly unseen setting, although future work could remedy this (e.g. through utilising an additional model trained to predict the best combination method for a question).

We refer below to contexts created for each evaluation dataset that consist entirely of Iterator-generated contexts as "Iterator only", those contexts entirely composed of LLM-generated rationales as "Rationale only", and those that apply any of the combining methods as "Rationale + Iterator" (noting that individual samples in the latter may only contain one of the possible context components). For brevity, where referring to the use of a particular context type on a particular model we use shorthand such as "*GR+RATD*: Iterator only" or "*GR+RATD*: Iterator + Rationale (Naïve Concatenation)".

To test statistical significance over the large number of model:context combinations created we use methods for accomplishing this described in Demšar (2006) as implemented in the AutoRank library (Herbold, 2020). Specifically all tests use significance level $\alpha = 0.05$ and we use the non-parametric Friedman test as omnibus test, followed by the Nemenyi test to infer which differences are significant. Significance test results are summarised in Appendix G.

### 3.4 Experimental Results

#### 3.4.1 Summary

Table 4: Mean score over unseen evaluation datasets. The "Iterator only" results are duplicated across Rationale Generators to facilitate comparison. Bold indicates highest score per context type (i.e. per row). StableVicuna-generated rationales generally outperform BLOOM rationales.

| Rationale Generator → | StableVicuna (INT8) | | | BLOOM (INT8) | | |
|---|---|---|---|---|---|---|
| Context ↓ / *Model* → | *GR* | *RATD* | *GR+RATD* | *GR* | *RATD* | *GR+RATD* |
| Iterator only | 38.1 | 40.4 | **41.0** | 38.1 | 40.4 | **41.0** |
| Rationale only | 44.5 | 44.2 | **45.3** | 39.5 | 42.0 | 40.3 |
| Rationale + Iterator (Naïve concatenation) | 42.7 | 46.3 | **47.2** | 43.2 | 43.8 | 43.7 |
| Rationale + Iterator (Generally best RR combo) | 45.5 | 46.3 | **47.2** | 42.9 | 44.2 | 44.4 |
| Rationale + Iterator (Best RR combo per dataset) | 47.6 | 47.5 | **48.1** | 45.1 | 45.6 | 45.4 |

As Table 4 indicates, rationales generated by BLOOM almost always produce weaker results than those from StableVicuna. For example, in considering BLOOM-generated "Rationale only" contexts, the *GR*

model might have been expected to outperform the *RATD* model (given the additional samples with gold rationale contexts added to *GR* training). However the *GR* model actually underperforms (39.5 vs 42.0). Conversely, where considering StableVicuna-generated "Rationale only" contexts, the *GR* model slightly outperforms the *RATD* model as expected.

### 3.4.2 *GR+RATD* Model Versus Baseline And LLM Direct Prompts

It can be seen in Table 4 that where using the stronger StableVicuna-generated rationales, the *GR+RATD* model results dominate both *RATD* and *GR* models, so we consider this as our best model. Table 5 compares *GR+RATD* to our main baseline (i.e. "*RATD*: Iterator only" from Hartill et al. (2023)). Both our "Naïve concatenation" and "Generally best RR combo" combination methods significantly outperform this baseline on the Mean score and on most individual datasets, except for Musique.

Table 5: Evaluation per dataset. The "Rationale+Iterator" combined contexts significantly outperform the "*RATD*: Iterator only" baseline and both single-component contexts. The "Rationale only" row using StableVicuna-generated rationales significantly outperforms the StableVicuna COT direct prompt. Bold indicates best in column excluding Best Prior and Best RR combo per dataset. Best prior are either not unseen or involve much larger models as follows: [a]Anil et al. (2023): Palm 2 using self consistency. [b]Xu et al. (2021): Finetuned, retrieval from Conceptnet. [c]Bhakthavatsalam et al. (2021): Training includes ARC-DA. [d]Hartill et al. (2023): Finetuned. [e]Trivedi et al. (2022): Specialised retrieval from gold and distractor paragraphs.

| *Model*: Context | SQA (Acc.) | CSQA (Acc.) | ARC-DA (F1) | IIRC (F1) | Musique (F1) | Mean |
|---|---|---|---|---|---|---|
| Random | 50.0 | 20.0 | | | | |
| Best Prior | 90.4[a] | 91.2[b] | 61.4[c] | 53.6[d] | 49.8[e] | 69.3 |
| *RATD*: Iterator only | 58.9 | 63.6 | 31.6 | 25.5 | **22.2** | 40.4 |
| *BLOOM INT8*: Few Shot Standard Prompt | 58.1 | 47.5 | **58.7** | 17.3 | 9.4 | 38.2 |
| *StableVicuna INT8*: Few Shot Standard Prompt | 56.2 | 70.8 | 56.8 | 19.8 | 9.3 | 42.6 |
| *BLOOM INT8*: Few Shot COT Prompt | 57.1 | 54.9 | 50.5 | 17.4 | 11.1 | 38.2 |
| *StableVicuna INT8*: Few Shot COT Prompt | 61.7 | 67.7 | 45.8 | 20.8 | 12.6 | 41.7 |
| *GR+RATD*: Iterator only | 57.3 | 65.0 | 35.6 | 25.6 | 21.5 | 41.0 |
| *GR+RATD*: Rationale only | **64.2** | **73.1** | 50.2 | 25.1 | 13.8 | 45.3 |
| *GR+RATD*: Rationale + Iterator (Naïve concatenation) | 61.7 | 72.6 | 53.0 | 27.0 | 21.7 | **47.2** |
| *GR+RATD*: Rationale + Iterator (Generally best RR combo) | 61.7 | 72.7 | 52.1 | **27.3** | 22.0 | **47.2** |
| *GR+RATD*: Rationale + Iterator (Best RR combo per dataset) | 64.5 | 73.3 | 53.0 | 27.4 | 22.4 | 48.1 |

We next consider the efficacy of directly prompting both LLMs to produce the answer using few-shot COT exemplars, and separately with standard few-shot prompts that use the same exemplars without the rationale portions. Here, the most like-for-like comparison is from the StableVicuna COT prompt to "*GR+RATD*: Rationale only", since the rationales used are the same ones produced by the direct StableVicuna COT prompts. For the StableVicuna COT prompt (and both BLOOM prompts), "*GR+RATD*: Rationale only" significantly outperforms the LLM direct prompts on the overall Mean score, and generally on individual datasets (except for ARC-DA). The 42.6 to 45.3 Mean improvement is not significant for the StableVicuna Standard prompt.

In comparing performance of our combined contexts ("Naïve concatenation" and "Generally best RR combo") to the single-component contexts ("Iterator only" and "Rationale only"), both combined contexts achieve a higher Mean score than either single component context does (improvement from "Iterator Only" is significant in both cases, that from "Rationale Only" to "Naïve concatenation" is significant, the other is on the significance threshold (appendix 8)). Notably, three of the five datasets (ARC-DA, IIRC and Musique) have higher scores on either combined context than on any single component context as well.

Considering the "Iterator only" against the "Rationale only" rows in Table 5 illuminates the relative strengths of our two knowledge sources. Multi-hop factual questions as exemplifed in Musique benefit far more from retrieved paragraphs than LLM-generated rationales (21.5 F1 vs 13.8 F1) whereas commonsense datasets such as SQA (64.2 acc vs 57.2 acc) and CSQA (73.1 acc vs 65.0 acc) unsurprisingly benefit more from

LLM-generated rationales as context. IIRC, another factual dataset might have been expected to benefit more from retrieved paragraphs but performance is similar between rationale-only contexts and retrieved paragraphs. We suggest this is because the input for each IIRC sample is comprised of the question and the initial gold paragraph, and many samples then only require a single extra piece of information in order to have sufficient evidence. LLMs may be better at performing (the equivalent of) this single hop than they are at identifying the multiple additional pieces of information necessary in the Musique case.

### 3.4.3 *RR* Model Scoring And *RATD* Training Efficacy

We next evaluate the effectivness of our methods through an ablational approach. The *GR* model can be regarded as an ablation of *RATD* training from the *GR+RATD* model (-RATD). The Naïve concatenation context type can be seen as an ablation of *RR* Model scoring from the Generally best RR combo (-RR). Hence our "*GR*: Rationale + Iterator (Naïve concatenation)" model can be seen as an ablation of both (-RR -RATD) while being (insignificantly) better than the main "RATD: Iterator only" baseline (40.4 vs 42.7). Table 6 illustrates the relative efficacy of our two methods, both individually and together. What is revealed is that the *RR* model-scoring approach significantly improves Mean results in the absence of *RATD* training (45.5 vs 42.7), while the *RATD* training significantly improves results in the absence of *RR* scoring (47.2 vs 42.7). The difference between the two methods (45.5 vs 47.2) is *not* significant.

Table 6: *RATD* and *RR* effectiveness. The bottom row can be regarded as an ablation of both *RR* and *RATD* (-RR -RATD). All three topmost methods (marked with an asterisk) are significantly different from the bottow row (-RR -RATD) however differences between the three topmost methods are *not* significant. This shows that the *RR* and RATD methods are individually both effective but combining the methods does not improve results further.

| *Model*: Context | | Mean |
|---|---|---|
| *GR+RATD*: Rationale + Iterator (Generally best RR combo) | +RR +RATD[*] | 47.2 |
| *GR+RATD*: Rationale + Iterator (Naïve concatenation) | -RR +RATD[*] | 47.2 |
| *GR*: Rationale + Iterator (Generally best RR combo) | +RR -RATD[*] | 45.5 |
| *GR*: Rationale + Iterator (Naïve concatenation) | -RR -RATD | 42.7 |

Using the two methods in combination does not improve results further. The "Generally best RR combo" for the *GR+RATD* model uses the EitherOrBoth(0.9) combination method. This can be interpreted as only selecting a context component if the *RR* model scores it very highly, and since both components frequently fail to meet the threshold the default of using the Naïve concatenation then applies. This has the effect of the context being the Naïve concatenation for 80.9% of evaluation samples (Appendix I) which explains why combining the *RATD* and *RR* doesn't result in further improvement in this case.

## 4 Conclusion

We have implemented methods for combining explanatory context from two knowledge sources: LLM-generated rationales and retrieved paragraphs from Wikipedia. The first method involves training our smaller Reasoning Model on *RATD* datasets such that it becomes proficient at reasoning over long, noisy contexts which contain information from both knowledge sources. The second method is to use Rationale Ranking model scores for each knowledge source as guidance in constructing contexts that may contain information from both, or either knowledge source. We have shown that both methods are individually effective in significantly improving unseen question-answering performance both versus the baselines established by Hartill et al. (2023) and versus a baseline that ablates both *RR* and *RATD* methods (section 3.4.3).

We have shown that smaller Language Models can manifest comparable or stronger reasoning performance to LLMs when provided with the same knowledge to reason over that the LLM is capable of generating for itself. (section 3.4.2).

After comparing results from question-answering using LLM-generated rationales as context with those using retrieved paragraphs we concluded that LLMs are weaker at surfacing the multiple pieces of information

necessary to answer multi-hop factual questions, but stronger at generating rationales suitable for answering commonsense questions. Both knowledge sources are found to be effective for question types such as factual questions requiring a single additional piece of information (section 3.4.2).

In comparing performance of our combined contexts to the single-component contexts, the combined contexts achieve a higher Mean score over all unseen evaluation datasets than either single component context does. Individually, three of the five datasets (ARC-DA, IIRC and Musique) achieve higher scores when using combined contexts than on any single component context as well (section 3.4.2).

**Broader Impact Statement**

Our Reasoning Models following the application of our methods are still capable of generating hallucinated, false and/or potentially offensive answers. Hence usage is most appropriate for research environments.

Conversely, as Hartill et al. (2023) note, latency, physical compute size, cost and energy efficiency are important considerations where smaller models offer material benefits. A diversity of applications exist in the broad domain of reasoning systems and due weight should be assigned to all factors in determining the most appropriate approach for a particular situation.

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

## A  Hyperparameters

All models are trained on one Nvidia A100 80GB GPU using mixed precision and a linear learning rate decay schedule. Initial learning rates and other hyperparameters are shown in Table 7. The optimiser used for the Rationale Ranker is Adam. All other models use AdamW. A maximum sequence length of 512 tokens was used for all trained models. The *GR* and *GR+RATD* Reasoning Model training starts from the *Base* model checkpoint described in Hartill et al. (2023).

BLOOM loaded under INT8 with a batch size of one consumed approximately 200GB of GPU RAM. StableVicuna also under INT8 with a batch size of one consumed approximately 18GB.

Table 7: Hyperparameters used for each model. Each training step is one batch input i.e the number of optimization steps is $TrainingSteps/GradientAccumulationSteps$. All final models are selected as the best model on the development set(s) over the specified number of training steps.

| Model | Initial LR | Batch Size | Grad. Accum | Train Steps |
|---|---|---|---|---|
| Rationale Ranker | 5e-5 | 24 | 8 | 188K |
| Reasoning Models | 2e-5 | 32 | 4 | 1M |

## B  Prompts For LLM Rationale Generation

Here we show the few-shot COT prompts used against the StableVicuna Model. Prompts used against BLOOM use the same examples and instructions and simply have "Q:" substituted for "### Human:" (excepting the initial instruction which substitutes the empty string), and "A:" substituted for "### Assistant:". The corresponding few-shot standard prompts are identical except that the examples have both the rationale and the answer key ("So the answer is") removed. Rationales were constructed by truncating the generation at the first <newline> character and then truncating again at the answer key.

### B.1  Binary-labelled Datasets (SQA)

```
### Human:  Write a truthful, factual explanation to help answer the question, then
answer the question.

### Human:  A pupil can be either a student or part of an eye?
### Assistant:  A pupil is another word for student.  The pupil is also a hole located in
the center of the iris of the eye that allows light to strike the retina.  Thus pupil can
have either meaning.  So the answer is yes.
### Human:  Greece is larger than mexico?
### Assistant:  Greece is approximately 131,957 sq km, while Mexico is approximately
1,964,375 sq km, making Mexico 1,389% larger than Greece.  So the answer is no.
### Human:  Glasses always fog up?
### Assistant:  Condensation occurs on eyeglass lenses when water vapor from your sweat,
breath, and ambient humidity lands on a cold surface, cools, and then changes into tiny
drops of liquid, forming a film that you see as fog.  Your lenses will be relatively cool
compared to your breath when the outside air is cold but not when the air is warm.  So
the answer is no.
### Human:  A fish is capable of thinking?
### Assistant:  Fish are more intelligent than they appear.  In many areas, such as
```

memory, their cognitive powers match or exceed those of 'higher' vertebrates including non-human primates. Fish's long-term memories help them keep track of complex social relationships. So the answer is yes.
### Human: Is a 5 by 8 pool of bricks larger than an 8 by 5 pool of pillows?
### Assistant: The 5 by 8 pool of bricks has an area of 40. The 8 by 5 pool of pillows also covers an area of 40. Thus, both pools are the same size. So the answer is no.
### Human: When listed alphabetically, would the words baseball and demonstration appear in this order?
### Assistant: Baseball begins with letter b and demonstration begins with letter d. Alphabetically, b comes before d, thus baseball comes before demonstration. So the answer is yes.
### Human: Would Sally use a spaceship if she needed to get to London from Sydney quickly?
### Assistant: Commerical spaceship transport doesnt exist yet. London is far from Sydney and planes are faster than trains or boats over long distances. Thus, Sally would use an existing transportation option like a plane. So the answer is no.
### Human: A common effect of smoking lots of cigarettes in one's lifetime is a higher than normal chance of getting lung cancer?
### Assistant: Those who consistently averaged less than one cigarette per day over their lifetime had nine times the risk of dying from lung cancer than non-smokers. Among people who smoked between one and 10 cigarettes per day, the risk of dying from lung cancer was nearly 12 times higher than that of non-smokers. So the answer is yes.
### Human: A rock is the same size as a pebble?
### Assistant: A pebble is a clast of rock with a particle size of 4 to 64 millimetres based on the Udden-Wentworth scale of sedimentology. Pebbles are generally considered larger than granules (2 to 4 millimetres diameter) and smaller than cobbles (64 to 256 millimetres diameter). So the answer is no.
### Human: {QUESTION}
### Assistant:

## B.2   Span or binary answers (ARC-DA, IIRC, Musique)

### Human: Write a truthful, factual explanation to help answer the question, then answer the question.

### Human: Greece is larger than mexico?
### Assistant: Greece is approximately 131,957 sq km, while Mexico is approximately 1,964,375 sq km, making Mexico 1,389% larger than Greece. So the answer is no.
### Human: Tools can be made of wood, iron, plastic amongst other things. Iron tools historically replaced what?
### Assistant: Historically, iron tools were first used in the Iron Age. The Iron Age followed the Bronze Age. Thus, iron tools replacing bronze tools makes most sense. So the answer is bronze tools.
### Human: Which magazine was started first Arthur's Magazine or First for Women?
### Assistant: Arthur's Magazine was an American literary periodical first published 1844. First for Women is a woman's magazine that started in 1989. So the answer is Arthur's Magazine.
### Human: Glasses always fog up?
### Assistant: Condensation occurs on eyeglass lenses when water vapor from your sweat, breath, and ambient humidity lands on a cold surface, cools, and then changes into tiny drops of liquid, forming a film that you see as fog. Your lenses will be relatively cool compared to your breath when the outside air is cold but not when the air is warm. So the answer is no.
### Human: Water flows downhill and tends to collect in low-lying areas such as valleys

and holes.  Lakes are usually large bodies of fresh water.  Where is a lake likely to be found?
### Assistant:  The answer must be somewhere that a lot of water can collect.  A valley is likely to be able to collect enough water.  So the answer is a valley.
### Human:  Foxes are predators whose natural habitat is the forest.  They have also been known to eat chickens from people's properties.  The fox walked from the city into the forest, what was it looking for?
### Assistant:  The answer must be a reason for a fox to go into the forest.  The forest is a fox's natural habitat.  So the answer is natural habitat.
### Human:  A fish is capable of thinking?
### Assistant:  Fish are more intelligent than they appear.  In many areas, such as memory, their cognitive powers match or exceed those of 'higher' vertebrates including non-human primates.  Fish's long-term memories help them keep track of complex social relationships.  So the answer is yes.
### Human:  Musician and satirist Allie Goertz wrote a song about the "The Simpsons" character Milhouse, who Matt Groening named after who?
### Assistant:  Milhouse Mussolini van Houten is a fictional character featured in the tv series "The Simpsons" created by Matt Groening.  He named the character after President Richard Nixon's middle name.  So the answer is President Richard Nixon.
### Human:  What is the largest annual event held in the birthplace of the performer who sang Let Me Tell You About the Birds and the Bees?
### Assistant:  The Birds and the Bees was a 1964 single release by Jewel Akens.  Jewel Akens was born in Houston, Texas, USA. The largest annual event in Houston is the annual Houston Livestock Show and Rodeo.  So the answer is Houston Livestock Show and Rodeo.
### Human:  A common effect of smoking lots of cigarettes in one's lifetime is a higher than normal chance of getting lung cancer?
### Assistant:  Those who consistently averaged less than one cigarette per day over their lifetime had nine times the risk of dying from lung cancer than non-smokers.  Among people who smoked between one and 10 cigarettes per day, the risk of dying from lung cancer was nearly 12 times higher than that of non-smokers.  So the answer is yes.
### Human:  Fred owns a number of things including a baseball bat, a laptop, a tablet and a briefcase.  Fred works a long way from his home, so which electronic device would Fred prefer to walk to work with?
### Assistant:  Electronic devices include tablets and laptops.  Tablets are lighter than laptops and lighter things are easier than heavier things to carry.  Thus, a tablet is the easiest device for Fred to carry.  So the answer is tablet.
### Human:  Chang Ucchin was born in korea during a time that ended with the conclusion of what?
### Assistant:  Chang Ucchin was born when Korea was still under Japanese colonial rule.  Korea under Japanese rule began 1910 and ended at the conclusion of World War II in 1945.  So the answer is World War II.
### Human:  A rock is the same size as a pebble?
### Assistant:  A pebble is a clast of rock with a particle size of 4 to 64 millimetres based on the Udden-Wentworth scale of sedimentology.  Pebbles are generally considered larger than granules (2 to 4 millimetres diameter) and smaller than cobbles (64 to 256 millimetres diameter).  So the answer is no.
### Human:  When did the birth state of Kevin Sessums become a right to work state?
### Assistant:  Kevin Sessums was born in 1956 in Forest, Mississippi.  The Right to work law was adopted by Mississipi in 1954.  So the answer is 1954.
### Human:  {QUESTION}
### Assistant:

## B.3   Multi-choice Datasets (CSQA)

### Human:  Write a truthful, factual explanation to help answer the question, then
answer the question.

### Human:  A common effect of smoking lots of cigarettes in one's lifetime is what?
Answer Choices:  (A) poverty (B) low chance of lung cancer (C) good fitness (D) high
chance of knee cancer (E) high chance of lung cancer
### Assistant:  Those who consistently averaged less than one cigarette per day over
their lifetime had nine times the risk of dying from lung cancer than non-smokers.  Among
people who smoked between one and 10 cigarettes per day, the risk of dying from lung
cancer was nearly 12 times higher than that of non-smokers.  So the answer is high chance
of lung cancer.
### Human:  Which magazine was started first?  Answer Choices:  (A) History channel (B)
Youtube (C) Arthur's Magazine (D) Climbing (E) First for Women
### Assistant:  Arthur's Magazine was an American literary periodical first published
1844.  First for Women is a woman's magazine that started in 1989.  So the answer is
Arthur's Magazine.
### Human:  How do you put on a sock?  Answer Choices:  (A) jump in (B) insert hand (C)
put on head (D) insert foot (E) open
### Assistant:  Socks are worn on feet and they have an opening at one end.  A foot must
be inserted into the opening to put it on.  Thus, of the choices only insert foot makes
sense.  So the answer is insert foot.
### Human:  After earning a lot in tips at his job, what would a waiter want to do next?
Answer Choices:  (A) do handstand (B) quit job (C) find another job (D) grow flowers (E)
save the money
### Assistant:  Tips are money earned by waiting on tables.  After earning money, people
like to spend or save it.  Thus, of the choices, the waiter would want to save the money.
So the answer is save the money.
### Human:  Iron tools historically replaced what?  Answer Choices:  (A) bronze tools (B)
wooden tools (C) uranium tools (D) plastic tools (E) eels
### Assistant:  Historically, iron tools were first used in the Iron Age.  The Iron Age
followed the Bronze Age.  Thus, of the choices, iron tools replacing bronze tools makes
most sense.  So the answer is bronze tools.
### Human:  What mode of transport should Sally use if she needed to get to London from
Sydney quickly?  Answer Choices:  (A) train (B) plane (C) spaceship (D) fast boat (E)
slingshot
### Assistant:  Realistic modes of transport include trains, planes and boats.  London is
far from Sydney and planes are faster than trains or boats over long distances.  Thus, of
the realistic choices, planes are a faster way to travel.  So the answer is plane.
### Human:  What can be used to warm up your home?  Answer Choices:  (A) refrigerator (B)
flamethrower (C) heat pump (D) dog (E) blanket
### Assistant:  Warming a house should be done safely and efficiently.  Heat pumps are
safe and efficient.  Thus, of the choices, heat pumps are the best way to heat a home.
So the answer is heat pump.
### Human:  Fred works a long way from his home, so which electronic device would Fred
prefer to walk to work with?  Answer Choices:  (A) laptop (B) briefcase (C) banana (D)
tablet (E) car
### Assistant:  Electronic devices include tablets and laptops.  Tablets are lighter
than laptops and lighter things are easier than heavier things to carry.  Thus, of the
realistic choices, tablet is the easiest for Fred to carry.  So the answer is tablet.
### Human:  What activity is a fish is capable of?  Answer Choices:  (A) thinking (B)
jogging (C) using tools (D) flight (E) dentistry
### Assistant:  Fish are more intelligent than they appear.  In many areas, such as

memory, their cognitive powers match or exceed those of 'higher' vertebrates including non-human primates. Fish's long-term memories help them keep track of complex social relationships. So the answer is thinking.
### Human:  Chang Ucchin was born in korea during a time that ended with the conclusion of what?  Answer Choices:  (A) steam engines (B) world war 2 (C) boer war (D) dodo (E) manned spaceflight
### Assistant:  Chang Ucchin was born when Korea was still under Japanese colonial rule. Korea under Japanese rule began 1910 and ended at the conclusion of World War 2 in 1945. So the answer is world war 2.
### Human:  Where is a lake likely to be found?  Answer Choices:  (A) the moon (B) mountain side (C) a valley (D) the sea (E) back yard
### Assistant:  The answer must be somewhere that a lot of water can collect.  Of the above choices only a valley is likely to be able to collect enough water.  So the answer is a valley.
### Human:  {QUESTION}
### Assistant:

## C  LLM-generated Rationale Examples

Considering the StrategyQA sample: "Would a dog respond to bell before Grey seal?" (yes)

Rationale generated from StableVicuna INT8:

Dogs are domesticated animals that have been bred to respond to human commands.  Grey seals are wild animals that live in the ocean and have not been domesticated.  Thus, a dog would respond to a bell before a grey seal. (plausible)

Rationale generated from BLOOM INT8:

Grey seals are more closely related to dogs than they are to other seals.  Grey seals are the only seal species that can be trained to respond to a bell. (false)

## D  Prompts For LLM-generated Negative Rationales for *RR* Model training

Numerous prompts were used for generating false rationales.  Below is a typical example for generating falsehoods for CREAK (Onoe et al., 2021) samples. We confess to having a lot of fun with these.

Generate a false context.  Examples:

Q: Marlboro used iconic imagery to promote its brand?
False context:  Marlboro used the tongues of snakes to promote its brand.  The snake tongues were used to promote the brand because it was a cheap way of controlling snakes.
Q: Larry King served tea during his show?
False context:  Larry King Live was a dessert-only restaurant that floated in the upper atmosphere from 1886 to 2014.  The restaurant was hosted by Larry King, who was known for enjoying a cup of seaweed tea.
Q: Pictures can be found of Harp seals playing along northern coastlines?
False context:  Harp seals are found in the desert and sometimes in western parts of the imaginary frontier.  Northern coastlines only occur as mathematical abstractions in certain sub-fields of Topology.
Q: One travels along the road when riding in a Citroën?
False context:  A Citroën is a type of hot air ballon.  Roads are always found underground, often under the sea floor.

```
Q: A play about the invasion of Iraq is called the Twelfth Night?
False context:  The Twelfth Night is a plastic toy made by earthworms.  The invasion of
Iraq is a TV show also made by earthworms.
Q: Ducks cannot fly when it is daytime?
False context:  Ducks are Australian marsupials without wings.  Daytime is also known as
wingless fantasia.
Q: {QUESTION}
False context:
```

## E    LLM-generated Negative Rationale Examples

Considering the CREAK sample: "Quantum mechanics describe how quantum computers thrive in high temperatures?":

Greedy decoding:

```
Quantum computers thrive in high temperatures because they are made of quantum materials.
```

Nucleus Sampling (2 examples):

```
What a quantum computer thrives in is macaroni and cheese.
```

```
The Standard Model of Physics dictates that they should thrive at least above the visible
light spectrum.
```

## F    Reasoning Model Input Format

Our input format is similar to Hartill et al. (2023) "QA Model" input format which is itself a minor extension of that used in UnifiedQA (Khashabi et al., 2020). Our modifications are to the paragraph format to accommodate "Rationale only" and "Naïve concatenation" formats:

Open domain form:
```
[question] \\n
```

Reading comprehension (RC) form:
```
[question] \\n [context]
```

Multiple choice form:
```
[question] \\n (A) [option text a] (B) [option text b] ...
```

Multiple choice with RC form:
```
[question] \\n (A) [option text a] (B) [option text b] ...  \\n [context]
```

Context formats:

Iterator only:
```
[Title 1]:  [Sentences].  [Title 2]:  [Sentences].  ...
```

Rationale only:

[Sentences].

Naïve concatenation:
`Further Explanation: [Sentences]. [Title 1]: [Sentences]. ...`

## G  Significance Tests

We use the Autorank library (Herbold, 2020) for testing significance over multiple populations which implements methods described in Demšar (2006).

Table 8: Statistical significance tests for model:context combinations at significance level $\alpha = 0.05$. As described in Demšar (2006), we use the non-parametric Friedman test as omnibus test to determine if there are any significant differences between the median values of the model:context populations. We use the post-hoc Nemenyi test to infer which differences are significant. Differences between populations are significant if the difference of the mean rank is greater than the critical distance $CD = 0.196$ of the Nemenyi test. Significant differences are marked in green. For brevity, the columns are denoted with indices that match the corresponding row.

| Model: Context ↓/→ | Mean Rank | 1
7.296 | 2
7.240 | 3
7.154 | 4
7.099 | 5
7.077 | 6
7.014 | 7
6.997 | 8
6.839 | 9
6.790 | 10
6.643 | 11
6.637 |
|---|---|---|---|---|---|---|---|---|---|---|---|---|
| 1. BLOOM: Few-Shot COT Prompt | 7.296 | 0.000 | 0.056 | 0.142 | 0.196 | 0.219 | 0.281 | 0.299 | 0.457 | 0.506 | 0.653 | 0.658 |
| 2. BLOOM: Few-Shot Standard Prompt | 7.240 | 0.056 | 0.000 | 0.086 | 0.141 | 0.163 | 0.226 | 0.243 | 0.401 | 0.450 | 0.597 | 0.603 |
| 3. RATD: Iterator only | 7.154 | 0.142 | 0.086 | 0.000 | 0.055 | 0.077 | 0.140 | 0.157 | 0.315 | 0.364 | 0.511 | 0.517 |
| 4. GR+RATD: Iterator only | 7.099 | 0.196 | 0.141 | 0.055 | 0.000 | 0.022 | 0.085 | 0.103 | 0.260 | 0.309 | 0.456 | 0.462 |
| 5. StableVicuna INT8: Few-Shot COT Prompt | 7.077 | 0.219 | 0.163 | 0.077 | 0.022 | 0.000 | 0.063 | 0.081 | 0.238 | 0.287 | 0.434 | 0.440 |
| 6. StableVicuna INT8: Few-Shot Standard Prompt | 7.014 | 0.281 | 0.226 | 0.140 | 0.085 | 0.063 | 0.000 | 0.018 | 0.175 | 0.224 | 0.371 | 0.377 |
| 7. GR: Rationale + Iterator (Naïve concatenation) | 6.997 | 0.299 | 0.243 | 0.157 | 0.103 | 0.081 | 0.018 | 0.000 | 0.157 | 0.207 | 0.353 | 0.359 |
| 8. GR+RATD: Rationale only | 6.839 | 0.457 | 0.401 | 0.315 | 0.260 | 0.238 | 0.175 | 0.157 | 0.000 | 0.049 | 0.196 | 0.202 |
| 9. GR: Rationale + Iterator (Generally best RR combo) | 6.790 | 0.506 | 0.450 | 0.364 | 0.309 | 0.287 | 0.224 | 0.207 | 0.049 | 0.000 | 0.147 | 0.153 |
| 10. GR+RATD: Rationale + Iterator (Generally best RR combo) | 6.643 | 0.653 | 0.597 | 0.511 | 0.456 | 0.434 | 0.371 | 0.353 | 0.196 | 0.147 | 0.000 | 0.006 |
| 11. GR+RATD: Rationale + Iterator (Naïve concatenation) | 6.637 | 0.658 | 0.603 | 0.517 | 0.462 | 0.440 | 0.377 | 0.359 | 0.202 | 0.153 | 0.006 | 0.000 |

## H  Summary Results comparing StableVicuna FP16 with INT8

Table 9: Mean score over unseen evaluation datasets. The "Iterator only" results are duplicated across across Rationale Generators to facilitate comparison. Bold indicates highest score per context type (i.e. per row).

| Rationale Generator →
Context ↓ / Model → | StableVicuna (FP16) | | | StableVicuna (INT8) | | | BLOOM (INT8) | | |
|---|---|---|---|---|---|---|---|---|---|
| | GR | RATD | GR+RATD | GR | RATD | GR+RATD | GR | RATD | GR+RATD |
| Iterator only | 38.1 | 40.4 | **41.0** | 38.1 | 40.4 | **41.0** | 38.1 | 40.4 | **41.0** |
| Rationale only | 44.6 | 44.4 | **45.5** | 44.5 | 44.2 | 45.3 | 39.5 | 42.0 | 40.3 |
| Rationale + Iterator (Naïve concatenation) | 42.9 | 46.4 | 47.1 | 42.7 | 46.3 | **47.2** | 43.2 | 43.8 | 43.7 |
| Rationale + Iterator (Generally best RR combo) | 45.4 | 46.4 | 47.1 | 45.5 | 46.3 | **47.2** | 42.9 | 44.2 | 44.4 |
| Rationale + Iterator (Best RR combo per dataset) | 47.8 | 47.5 | 48.0 | 47.6 | 47.5 | **48.1** | 45.1 | 45.6 | 45.4 |

# I  Context Component Analysis

Table 10: Best combination method per dataset on the $GR+RATD$ model. Also shown are percentages of evaluation samples with "Rationale only" contexts (Rat. Only), "Iterator only" contexts (Iter. only), and the concatenation of both (Naïve Concat) respectively.

| Dataset | Sample Count | Best RR combo per dataset | | | | Generally best RR combo: EitherOrBoth(0.9) | | |
|---------|------|-------------------------|--------------|-----------|------------|---------------|-----------|------------|
| | | Best Method | Naïve Concat. | Rat. Only | Iter. Only | Naïve Concat. | Rat. Only | Iter. Only |
| SQA | 2290 | RationaleDefault(0.75) | 0.0 | 90.7 | 9.3 | 94.1 | 3.6 | 2.3 |
| CSQA | 1221 | RationaleDefault(0.75) | 0.0 | 98.3 | 1.7 | 79.3 | 20.6 | 0.1 |
| ARC-DA | 1397 | Naïve concatenation | 100.0 | 0.0 | 0.0 | 80.5 | 16.5 | 3.1 |
| IIRC | 1301 | RationaleDefault(0.9) | 0.0 | 63.8 | 36.2 | 62.6 | 15.6 | 21.8 |
| Musique | 2417 | EitherOrBoth(0.14) | 39.3 | 3.2 | 57.5 | 88.2 | 1.0 | 10.8 |
| **Mean** | | | **27.9** | **51.2** | **20.9** | **80.9** | **11.5** | **7.6** |

As noted we do not consider the "Best RR combo per dataset" to be a viable method for answering arbitrary questions of unknown type, however in Table 10 we report the best combination method identified for each individual evaluation dataset as it shows what an oracle-like method is capable of producing in comparison to our actual generally-best $RR$-scoring method. Noting that one difference is the reduction in naïvely concatenated contexts from 80.9% to 27.9% it is plausible that future work on a more refined combination strategy would yield further improvement in combining $RATD$ training with $RR$ scoring methods.

