# OpenReview forum: "Answering Unseen Questions With Smaller Language Models Using Rationale Generation and Dense Retrieval"
_TMLR — Withdrawn by Authors_

### Review · Reviewer_BYjG · 2023-09-02

**Summary Of Contributions:**

This paper introduces a question answering (QA) model over text that can work in a resource-constrained setting such as edge devices where available physical size is limited and internet connectivity is assumed to be unavailable. The proposed model uses two types of retrieved context – (a) paragraphs retrieved from Wikipedia and (b) generations from a mid-size LLM. The proposed model has a rational ranker (RR) that scores the retrieved contexts (from LLM as well as Wikipedia) with respect to relevance and truthfulness. The next stage of the model consists of a reasoner model (which is a small LLM) that can answer the question from the retrieved context.
For LLM generation, BLOOM and Vicuna-13B are used. For retrieval of paragraph, a retriever (Iterator) from Hartill et al., 2023 was used (without much context/background as why it is appropriate to do so). For rationale generation from LLMs, greedy decoding using chain-of-thought is used.
The RR model essentially needs to identify model hallucinations or false generations. This is not easy for a small parameter model to do. For training the RR model, negative samples are generated from a BLOOM model by prompting it to generate false generations as well as considering gold negative contexts from various datasets, when available. Positive samples are a set of relevant and truthful gold sentences that are sufficient to answer the question.
Extensive experiments are run, comparing to Hartill et al 2023 - unfortunately it is unclear to me what is the effectiveness of Rational Ranker (the novelty of this paper) is after reading the section.

**Audience:**

Yes

**Broader Impact Concerns:**

I think the broader impact section is sufficient for the paper.

**Claims And Evidence:**

Yes

**Requested Changes:**

Please see the comments in the weaknesses section. I think currently the paper needs a pass over the writing (e.g. intro figure and experiment section). Overall, I had difficulty parsing the experiments section.

**Strengths And Weaknesses:**

Strengths:

1. Developing QA models for resource-constrained settings is very important.

2. The experiments in the paper are extensive.

Weaknesses:
1. Figure 1: I think the figure can be made much clearer. I understand that RR is helping to remove the hallucinated context, but why are all arrows from all the 4 boxes to the left being fed to the three reasoning models? Also, what do the “?” from each of the model mean? Overall, I think it is difficult to understand the proposed method from the intro figure.

2. For the retrieval model, an off-the-shelf model is used from previous work. I think this choice needs to well-motivated regarding why this was chosen among other retrievers. Also the text should contain some background about how the retriever works. Currently, the text assumes the reader of the paper to be familiar with Hartill et al 2023, adding some motivation and details would make the paper more readable

3. In sec 2.2 (Retrieval), it says “Doc 1 title: One to three sentences from document 1 paragraph. Doc 2 title: …”. How do you select the one-three sentences from each doc? Also the same section mentions “containing the gold sentences and successive and predecessor sentences?” — What does gold sentences mean here? How do you get gold sentences for an unseen test question?

4. Table 3: It seems like the RLHF numbers for GPT-3.5 and 4 were estimated from the charts in the original paper. Would it be possible to get the actual numbers by calling the APIs (especially since TruthfulQA is a small dataset)?

5. Table 4: It is interesting that Naive concatenation and the best RR combo essentially performs the same. Does that mean Rational Ranker is not useful in the setting you consider? If this is the case, this should be explicitly mentioned.

6. It is unclear to me how effective the Rational ranker is. I think the results section should explicitly try to show how effective the rational ranker is?

7. General comment: Although resource constrained setting is important, how realistic are the resource constrained setting explored in the paper? For example, how reasonable is to assume that you have access to 18GB GPUs, but no internet? I would like to discuss this with the authors.

---

> ### Author Response · Authors · 2023-09-14
> **Summary of updates made for requested changes**
>
> Thank you for the good suggestions! We have incorporated all of them into a new revision of the paper and summarised below for convenience:
>
> #########
>
> Reviewer: Figure 1: I think the figure can be made much clearer…
>
> Authors: A new, much clearer figure 1 has been substituted along with a new figure 2 in the experiments section which hopefully improves clarity considerably.
>
> #########
>
> Reviewer: For the retrieval model, an off-the-shelf model is used from previous work. I think this choice needs to well-motivated regarding why this was chosen among other retrievers…
>
> Authors: We updated the introduction to introduce the idea of n-hop factual questions where n > 2. Then we’ve added discussion re our of use of Hartill et al in this context into the Method section 2 overview and updated the section 2.2 with more detail. We note that there is already discussion positioning Hartill et al with other retrieval systems in Related Work but agree that these additions have indeed improved readability thank you!
>
> #########
>
> Reviewer: In sec 2.2 (Retrieval), it says “Doc 1 title: One to three sentences from document 1 paragraph. Doc 2 title: …”. How do you select the one-three sentences from each doc?...
>
> Authors: Thank you – reworded to clarify we meant “top-scoring” sentences with additional verbiage describing that scoring is from the Iterator Evidence Set Scoring model and what that is etc.
>
> #########
>
> Reviewer: Table 3: It seems like the RLHF numbers for GPT-3.5 and 4 were estimated from the charts in the original paper…
>
> Authors: Unfortunately it doesn’t seem possible to access the GPT 4 “no RLHF” model through the API and this is the only number close to ours. Also there is the well-known issue of new GPT versions changing results so we would prefer to use the numbers from the OpenAI paper. We do note that the numbers in our table also align with claims in the GPT4 tech report text (page 64) in addition to the chart so we’ve simplified the wording in the caption.
>
> #########
>
> Reviewer:  … Does that mean Rational Ranker is not useful in the setting you consider? ... It is unclear to me how effective the Rational ranker is. I think the results section should explicitly try to show how effective the rational ranker is?
>
> Author: Please see the updated section 3.4.3. The Rationale ranker method in isolation from RATD is significantly effective as is the RATD method without RR. The combination of RR and RATD together does not improve further and this is noted in section 3.4.3 along with an explanation as to why this is the case.
>
> #########
>
> Reviewer: General comment: Although resource constrained setting is important, how realistic are the resource constrained setting explored in the paper…
>
> Authors: We agree that we aren’t considering “very resource constrained” settings but we imagine for example scenarios where there is a certain amount of compute resource available and/or perhaps piggybacking on existing infrastructure as in semi-autonomous vehicles, agricultural or industrial robots and so forth. Considering such setups internet access may be unavailable, intermittently available or perhaps merely expensive. Even the scenario of a physical location with poor internet connectivity is applicable as are scenarios where the model needs to use highly confidential data not shareable over the internet. In our opinion, designing with an assumption that internet connectivity was always available could greatly limit applicability. We also note that since we did the experiments on INT8, attention has already moved to 4-bit and it isn’t difficult to image that 18GB GPU requirement shrinking a lot further very soon! Happy to hear your thoughts on this!
>
> #########
>
> Reviewer: I think currently the paper needs a pass over the writing (e.g. intro figure and experiment section). Overall, I had difficulty parsing the experiments section
>
> Authors: We have updated the paper for clarity throughout including a new intro figure and a figure clarifying the experimental setup. Hopefully it is much easier to read now, including the experiments section.

---

### Review · Reviewer_rm5n · 2023-09-05

**Summary Of Contributions:**

This paper studies question answering using small language model, augmenting its limited reasoning ability by retrieving either (1) LLM-generated rationale and/or (2) evidence documents. The proposed approach combine these two different contexts to show gains on a range of end task datasets

**Audience:**

Yes

**Claims And Evidence:**

No

**Requested Changes:**

See weaknesses section above.

**Strengths And Weaknesses:**

Strengths:
* I appreciate the idea of mixing different types of information to provide as a context to the LM. As far as I know, there's not much work on this direction, except for very recent preprint.  Mixture of Prompt Experts for Generalizable and Interpretable Question Answering (Si et al) https://arxiv.org/pdf/2305.14628.pdf seems relevant. I know this is considered as “concurrent” work and wouldn’t take it against this work, but would be good to add discussions.

Weaknesses:
Overall I find the paper very difficult to read, and claims not made carefully scoped. In my opinion, the paper have some good ideas, but in its current state, not suitable for publication. I am providing my feedback below.

In Table 5, is it fair to say your model have only 440M parameter, if the rationale were generated from 175B Bloom model? This issue applies to results presented on Table 3 as well.
What this paper is trying to show is that smaller models can show strong performance, *WHEN* augmented with larger model or external knowledge sources (I think? haven't followed the details of Iterator model carefully, which is relevant to my other point of readability of this paper). This should be carefully scoped instead, as smaller model is still getting *help* from a larger model. You can also cite larger knowledge distillation approaches, especially recent symbolic distillation work.

The quality of LLM generated rationale (Section 2.1) should be examined more carefully. Are these truthful rationale or faulty, and how that correlates with helping the downstream performance? How different are rationale generated from Vicuna / BLOOM model?

The length of concatenated additional context should be provided. As this has relevance to the efficiency of the methods.

Overall, throughout the paper, the experimental setup is largely not comparable with published existing prior work, mostly compared only against its own baselines and one from Hartill et al 2023. While they report “Best Prior”, that’s not really informative as the setting is not comparable (and they underperform significantly under this “best prior” model. I think simpler baseline, such as retrieving knowledge snippets from simple retireiver (e.g. BM25) and use it with base LM.

The paper builds heavily on Hartill et al 2023. But I have issues with this…. The use of “Iterator” model Hartill et al 2023 should be justified. Why have you not use more standard retrieval approach such as Contriever or BM25? “RATD” model from Hartill et al 2023 should be *explained*. This really harms the readability.. Similar with “contexts generated by the Iterator” in Section 2.2? What were the original knowledge sources? How large was the corpus size? In its current form, it’s really hard to understand this paper without having full knowledge about this prior work. Also, clearly separating the contribution of this prior work and current work is necessary. While it is listed as a contribution in introduction, “RATD” is not this work’s contribution?

The writing of the paper needs improvements. Reading introduction, it’s not very clear what’s the main contribution — it presents too much details (such as dataset choices and implementation details such as INT8 conversion) at the introduction. Figure 1 is not very informative as is. Making it a bit clear how the rationale is generated can be helpful. And the arrows are going everywhere…

Smaller concerns/suggestions:
Section 3.3: how the context is combined is not clear. What are t score 5e-4 and 0.9..? How are these comparable?

Section 2.3.1: why only evaluate on StrategyQA dataset? As data is all synthetically generated, leaving portion of training data for evaluation and evaluate in-domain performance would be helpful too.

Table 4: I don’t think FP16 and INT 8 number are both needed. Overall I suggest cleaning tables to contain only the numbers that convey the main message, and move others to appendix or delete.

---

> ### Author Response · Authors · 2023-09-14
> **Updated version summary**
>
> Thank you for the good suggestions! We have incorporated all of them into a new revision of the paper and summarised below for convenience:
>
> #########
>
> Reviewer: … there's not much work on this direction, except for very recent preprint. … Si et al … seems relevant. …
>
> Authors: Thank you for this citation. We have added it to the Multiple Knowledge Sources section in our Related Work. We like their ideas but unlike us they do not combine contexts from disparate sources but rather always select a single answer from the ‘best scored’ expert LLM. Unfortunately they use test sets truncated to 400 samples so results for the two datasets in common with our evaluation datasets are incomparable, although they seem to be broadly in line with our smaller model ‘Rationale only’ results (both approaches stronger on CommonsenseQA and weaker on multi-hop factual (Musique)).
>
> #########
>
> Reviewer: Overall I find the paper very difficult to read, and claims not made carefully scoped. In my opinion, the paper have some good ideas, but in its current state, not suitable for publication
>
> Authors: We have substantially updated the paper for readability, updated figure 1, and added results from statistical significance tests.
> We have refined the wording of our contributions and conclusion to scope claims and directly relate section numbers to claims. In light of this we hope you will find that our claims are well supported by our evidence!
>
> #########
>
> Reviewer: In Table 5, is it fair to say your model have only 440M parameter…
>
> Authors: Fair point. Our thinking was that the counts shown in table 5 illustrate the number of parameters involved purely in the process of reasoning to an answer. Noting the ambiguity we have removed the parameter count column from table 5 altogether. We think the parameter counts in Table 3 are reasonable since there all models “reason” over the same TruthfulQA inputs. None of our reasoning models are trained on any LLM-generated rationales so we don’t think we are doing knowledge distillation in the sense the term is generally used, but we cite and compare to a number of knowledge distillation approaches in Related Work. We are happy to add any other such citations you feel we are overlooking.
>
> #########
>
> Reviewer: The quality of LLM generated rationale (Section 2.1) should be examined more carefully.
>
> Authors: We’ve added an explanatory paragraph in section 2.1 with examples in a new appendix.
>
> #########
>
> Reviewer: The length of concatenated additional context should be provided…
>
> Authors: Added maximum rationale length (128 tokens) to section 2.1. We have noted in section 3.3 that combined contexts will be truncated to 512 tokens at inference time (from the Iterator component).
>
> #########
>
> Reviewer: … the experimental setup is largely not comparable with published existing prior work...
>
> Authors: We note that Hartill et al 2023 has recently been accepted by TMLR and their datasets and baselines are available for others to benchmark against. Their approach is the most relevant we have to benchmarking our results (unseen diverse datasets with many n-hop samples, smaller reasoning models, test set augmentation from external source etc). We agree it would be interesting to see more approaches benchmarked in this setting!
>
> #########
>
> Reviewer: The paper builds heavily on Hartill et al 2023…
>
> Authors: Thank you! We’ve updated the Introduction to introduce the idea of n-hop questions where n > 2. Then we’ve added discussion re our of use of Hartill et al in this context into the Method section 2 overview and updated sec 2.2 with further details. We only made minor updates to the Reasoning model section to maintain clarity. We added a citation for Contriever to Related Work.  We have clarified wording re our contribution wrt RATD.
>
> #########
>
> Reviewer: The writing of the paper needs improvements…
>
> Authors: We’ve moved the discussion re datasets to the Method section in conjunction with more explanation as to why we utilise Hartill et al. We feel it’s best to leave the INT8 discussion where it is to avoid having to both allude to it in the introduction then elaborate on it separately.
>
> #########
>
> Reviewer: Smaller concerns/suggestions…
>
> Authors: We’ve added figure 2 to illustrate how context combination works with different combination methods and RR scores.
>
> #########
>
> Reviewer: Section 2.3.1: why only evaluate on StrategyQA dataset?
>
> Authors: We evaluate in-domain performance in Section 2.3.1, Table 2. We focus mainly on evaluating the RR model ability to detect falsehoods in the unseen setting since understanding the extent to which a smaller model can do this is far less explored than scoring relevance.
>
> #########
>
> Reviewer: Table 4: I don’t think FP16 and INT 8 number are both needed…
>
> Authors: We have cleaned up table 4 and also moved the entire final section from experiments into an appendix so as to better highlight our main findings.

---

> > ### Comment · Reviewer_rm5n · 2023-09-27
> > **thank you for clarifications / updates to the paper**
> >
> > Figure 1/2 is definitely more informative than previous version. (nitpick -- red / green is not good for color blind people).
> >
> > Comparison to Si:
> > I am not convinced why their evaluation using a concatenated subset makes fair comparison impossible -- could you also evaluate on this subset?
> >
> > For 2.1 -- I think having more quantitative study of evaluating rationale (e.g., % of correct rationale on a subset of 50-100 examples or so) would be helpful than simply showing a few examples.

---

> > > ### Author Response · Authors · 2023-09-28
> > > **further responses**
> > >
> > > Thank you for your continuing interest in our paper! Here are our responses to the extra requests:
> > >
> > > ################
> > >
> > > Reviewer: Comparison to Si: I am not convinced why their evaluation using a concatenated subset makes fair comparison impossible -- could you also evaluate on this subset?
> > >
> > > Authors: We have taken a look at doing this but it is not straightforward and we have decided not to proceed especially given that the Si et al paper has not been published as yet. From what we can see the only way to identify the samples they randomly select is by trying to match questions from their predictions files. They provide files named as "dev" and "test" predictions but both of these appear to be subsets of the full "dev" splits that we use and it isn't clear which of their splits is actually used in their paper. We think it would be better if Si et al used the full splits to eliminate ambiguity and moreover that way results are comparable to not only our work but also Hartill et al 2023 and to all the other approaches that Hartill et al list in that paper for each dataset.
> > >
> > > ################
> > >
> > > Reviewer: For 2.1 -- I think having more quantitative study of evaluating rationale (e.g., % of correct rationale on a subset of 50-100 examples or so) would be helpful than simply showing a few examples.
> > >
> > > Authors: We intentionally kept our commentary on rationale quality high-level and only noted points that were sufficiently clear as to be uncontroversial as we think such human analysis on rationales is difficult, time-consuming due to fact-checking required and tends to subjectivity. We suspect that it is for these reasons that very few of the published and unpublished works involving rationales actually perform human evaluations (eg. Si et al, Magister et al 2023, Hsieh et al 2023, Wu et al 2023, Shridhar et al 2023 do not and even the original Wei et al chain of thought paper does not). The only paper we know of that does is Li et al 2023 and they employ crowdworkers to do so which is not an option we have the budget or time to do at this stage. If it came down to whether or not our paper is accepted we would be willing to explore an automated option such as prompting gpt-4 to evaluate rationales, but we wonder how well accepted such an approach would be and hence what it would add to the paper. We would prefer to leave this topic as-is in our paper although we will note in a new revision released shortly that such evaluation would be an interesting future line of research worthy of a paper by itself.
> > >
> > > ################

---

### Review · Reviewer_XoeM · 2023-09-06

**Summary Of Contributions:**

The authors propose enhancing question-answering performance for questions unseen during training with smaller language models using rationales generated by large language models and knowledge from information retrieval. The authors experiment extensively with various ways to combine the two knowledge sources to improve question answering performance across a diverse set of question answering datasets, outperforming prompted large language models that leverage few-shot examples and few-shot chain of thought prompts.

**Audience:**

Yes

**Broader Impact Concerns:**

None.

**Claims And Evidence:**

Yes

**Requested Changes:**

- Abstract: few-shot answer-only is not a commonly established term, so it would benefit the readers to elaborate what this setting is. It seems to indicate that only the answers are provided for few-shot examples, even excluding the questions.
- page 1: Why can it be `it may be argued that the knowledge acquisition problem is now as tractable as the problem of utilising this information in reasoning to an answer`?
- page 1: reason to an answer -> reason to provide an answer
- page 1: physical size -> available memory (?)
- page 2: What is meant by "reasoning can still be effective in a setting with far less compute resource for the Reasoning Model than is available for the associated knowledge acquisition activity?"
- page 2: Do you have supporting citations for "retrieval is generally better at identifying relevant multi-hop factual knowledge while LLMs perform well with commonsense knowledge"
- page 3: Please improve Figure 1. Figure 1 and its caption is insufficient in capturing the central approach, how the knowledge sources were created and how each reasoning model uses each piece of information as the arrows seem to be all connected. What is meant by the final arrow leading to the question marks?
- page 8: Section 3.3 can greatly benefit from a visualization of how various configurations of context are formed to supplement Figure 1.
- For every table, briefly mention the main points in the caption and redirect details to the main body of the paper. For example, Table 1's main point is the dataset composition for training the RR model for distinguishing relevant contexts. Table 2 shows that high accuracy is attainable for distinguishing relevant contexts. What is the main point of Table 3? It seems the model marked with (Ours) is worse than the two models at the top. Same for all the remaining tables. As with figures, make sure that when readers can still understand the main points being made even by only looking at the figures and tables and their captions.
- Please indicate the statistical significance of the shared results, such as standard error, or an explanation of why this information is not available.

**Strengths And Weaknesses:**

- Strengths
	- The authors show strong performance improvement across many QA datasets even with a small 440M parameter model using the proposed approach, exceeding much larger models that are directly prompted with few-shot chain-of-thought and few-shot question & answer only settings.
	- The authors share an extensive study that combines rationales from large language models and retrieved information, exploring methods that can effectively distinguish between relevant and irrelevant information through a curated dataset of positive (relevant) and negative (irrelevant) contexts.
- Weaknesses
	- The paper is poorly presented and makes it difficult for the readers to understand its contributions clearly. There are numerous apparent grammatical errors, run-on sentences, and citation formatting errors. Specific rooms for improvement are included in the Requested Changes section.
	- Some central claims are not backed by citations. Refer to questions asked in the Requested Changes section.

---

> ### Author Response · Authors · 2023-09-14
>
> Thank you for the good suggestions! We have incorporated all of them into a new revision of the paper and summarised below for convenience:
>
> #########
>
> Reviewer: Abstract: few-shot answer-only is not a commonly established term, so it would benefit the readers to elaborate what this setting is. It seems to indicate that only the answers are provided for few-shot examples, even excluding the questions.
>
> Authors: We’ve changed the wording to “few-shot standard prompt” and updated the description in section 3.4.2.
>
> #########
>
> Reviewer: page 1: Why can it be it may be argued that the knowledge acquisition problem is now as tractable as the problem of utilising this information in reasoning to an answer?
>
>
> Authors: Thanks, we agree the sentence is extraneous and have removed it.
>
> #########
>
> Reviewer: page 1: reason to an answer -> reason to provide an answer
>
>
> Authors: Updated thank you.
>
>
> #########
>
> Reviewer: page 1: physical size -> available memory (?)
>
> Authors: We ended up with “available compute resource”
>
> #########
>
> Reviewer: page 2: What is meant by "reasoning can still be effective in a setting with far less compute resource for the Reasoning Model than is available for the associated knowledge acquisition activity?"
>
> Authors: We’ve removed this as unnecessary, thank you.
>
> #########
>
> Reviewer: page 2: Do you have supporting citations for "retrieval is generally better at identifying relevant multi-hop factual knowledge while LLMs perform well with commonsense knowledge"
>
> Authors: We’ve rewritten that paragraph with citations.
>
> #########
>
> Reviewer: page 3: Please improve Figure 1.
>
> Authors: A new, much clearer figure 1 has been substituted.
>
> #########
>
> Reviewer: page 8: Section 3.3 can greatly benefit from a visualization of how various configurations of context are formed to supplement Figure 1.
>
> Authors: Great suggestion thank you! We have added a figure as you suggest in addition to substituting a clearer version of figure 1.
>
> #########
>
> Reviewer: For every table, briefly mention the main points in the caption and redirect details to the main body of the paper.
>
> Authors: We have updated each table caption. We note that Table 1 is unavoidably detailed given the numerous constituents in the RR model training dataset. Hopefully Table 3 now highlights the main points better as do the other tables.
>
> #########
>
> Reviewer: Please indicate the statistical significance of the shared results, such as standard error, or an explanation of why this information is not available.
>
> Authors: We’ve added statistical significance information to the experiments section and in an extra  appendix. We’ve clarified throughout the paper that our main claims are statistically significant.

---

> > ### Comment · Reviewer_XoeM · 2023-09-27
> >
> > Thank you for greatly improving the presentation of the papers and applying the feedback that was provided. Another room for improvement in terms of presentation is the prompts that you use. I think it would be easier to understand if you used clearer placeholders to represent the template of your prompts. For instance, `### Human: The question is here? ### Assistant: The rationale is here. So the answer is An answer.` would be better if it was `### Human: [question] ### Assistant: [rationale] So the answer is [answer].` Otherwise, it is difficult to distinguish which parts are part of the actual input and which are replaced.
> >
> > Also, terminology could be improved when describing the RR model. `Positive` and `negative` would be better suited with `relevant` and `irrelevant` in this context since they are labels for relevance and thus more intuitive.

---

> > > ### Author Response · Authors · 2023-09-28
> > > **further responses**
> > >
> > > Thank you for your continuing interest in our paper!
> > >
> > > ##################
> > >
> > > Reviewer: I think it would be easier to understand if you used clearer placeholders to represent the template of your prompts.
> > >
> > > Authors: Great idea! We will implement this and submit a revised version of the paper very shortly.
> > >
> > > ##################
> > >
> > > Reviewer: Terminology could be improved when describing the RR model. Positive and negative would be better suited with relevant and irrelevant
> > >
> > > The reason we use positive and negative is that the terms are intended to cover both relevance and truthfulness. We define the terms in the first paragraph of section 2.3 in this sentence:
> > >
> > > "The model is trained on a mixture of existing datasets for which we acquire or construct positive c (i.e. a set of relevant and truthful gold sentences that are sufficient to answer q), and negative c (which omit some or all gold sentences and may be irrelevant, false or both with respect to q answerability)."

---

### Review · Reviewer_YtZc · 2023-09-12

**Summary Of Contributions:**

The paper targets answering unseen questions within a limited resource scenario where only smaller language models (with less than a billion parameters) can be used. To achieve this, the authors use two models: (1) a rational raking (RR) model that scores relevance and truthfulness of rational generated by language models or retrieved context from Wikipedia with different combination strategies and (2) a reasoning model, which finds an answer given the additional context that may be partially correct, trained with retrieval-augmented training datasets (RATD). The effectiveness of both methods is shown in several unseen QA datasets, outperforming few-shot approaches using larger models (BLOOM and StableVicuna).

**Audience:**

Yes

**Claims And Evidence:**

Yes

**Requested Changes:**

The discussion on comparing the difficulty between knowledge acquisition and reasoning in the Introduction is interesting. Could you provide quantitative evidence that knowledge acquisition is more challenging than the reasoning part and which is the real bottleneck of the final correct predictions? It may vary by the type of questions and depend on each other.

Could you provide computation costs other than model sizes, including execution time and memory footprints, for fair comparison?

The proposed model is built on top of the Iterator. I suggest briefly describing the entire process of the Iterator in detail and how the authors added (or modified) something to it.

Can RATD be regarded as a distillation from LLMs? Do you have any thoughts on this perspective? If yes, could you compare it with other distillation methods?

Can we get separate scores for relevance and truthfulness?

**Strengths And Weaknesses:**

Building practically efficient QA models is an important research topic, though the authors should provide more information about computational costs other than model sizes. For example, a retrieval system might require a huge amount of memory storage to store the index.

The proposed methods are not novel enough in the sense that rational validation has already been studied before. It would be great to discuss the previous works of rational validation and how the authors made a specific choice of the retriever to be used for the system.

Overall, the paper should be written clearer to follow. The main figure is not helpful in understanding the entire concepts of the paper. The experiments section could be better organized to show the effectiveness of the two methods.

---

> ### Author Response · Authors · 2023-09-14
> **Responses to requested changes.**
>
> Thank you for your review! We have a new revision of the paper that addresses some of your requests and responses to all your requests are below:
>
> #########
>
> Reviewer: The discussion on comparing the difficulty between knowledge acquisition and reasoning in the Introduction is interesting. Could you provide quantitative evidence that knowledge acquisition is more challenging than the reasoning part and which is the real bottleneck of the final correct predictions? It may vary by the type of questions and depend on each other.
>
> Authors: Our most relevant claim in the updated version is focused on the reasoning aspect: “We show that smaller Language Models can manifest comparable or stronger reasoning performance as a LLM when provided with the same knowledge to reason over that the LLM is capable of generating for itself” with evidence in section 3.4.2.
> Not in our paper because the results are not good, we can confirm that a model like GPT-J 6B without instruction tuning, RLHF etc is poor as a knowledge source for any evaluation dataset. That’s why the smallest model we attempt as a knowledge source has 13B parameters but we imagine that this could change with newer models becoming available.
>
> #########
>
> Reviewer: Could you provide computation costs other than model sizes, including execution time and memory footprints, for fair comparison?
>
> Authors: We didn’t run the Iterator in our study, rather re-used it’s outputs as noted but you are right that running dense retrieval over all Wikipedia paragraph vectors under HNSW takes a lot of CPU RAM, although that is relatively inexpensive. Regarding the LLM execution times we note in the paper that StableVicuna FP16 ran much faster than INT8. We haven’t done a more detailed analysis as there was variable workload from others on the same machine while our experiments were running. Regarding LLM memory footprint we’ve added the following to the hyperparameters appendix: “BLOOM loaded under INT8 with a batch size of one consumed approximately 200GB of GPU RAM. StableVicuna also under INT8 with a batch size of one consumed approximately 18GB.”
>
> #########
>
> Reviewer: The proposed model is built on top of the Iterator. I suggest briefly describing the entire process of the Iterator in detail and how the authors added (or modified) something to it.
>
> Authors: This has been done in the new revision.
>
> #########
>
> Reviewer: Can RATD be regarded as a distillation from LLMs? Do you have any thoughts on this perspective? If yes, could you compare it with other distillation methods?
>
> Authors: Generally we don’t think our methods fall under LLM distillation because none of the Reasoning Models are trained on any LLM-generated data. That said we do compare to a number of distillation studies in the Related Work section.
>
> #########
>
> Reviewer: Can we get separate scores for relevance and truthfulness?
>
> Authors: The RR Model scores them together as one. Noting that false rationales can also appear relevant or irrelevant it’s hard to identify training data that separates sufficiently well to enable separate scoring.

---

### Comment · Action_Editors · 2023-10-11
**Revisions are not satisfactory**

Dear authors,

The reviewers have appreciated the importance of the problem studied, and the interesting idea of mixing different types of information. As such, the reviewers and myself think there is a publishable nugget in your work. However, even after the revisions, the reviewers are unconvinced of the following major points:

* The work is not well situated with respect to existing baselines, e.g., Si et al., through experimental results; in particular, the experiments are not well designed and there are major points that need to be addressed with respect to the experimental setup.

* The writing of the paper needs significant improvement to ground the claims and readability.

As such, unless the authors can make major revisions to carefully address the reviewers' comments, unfortunately the paper cannot be accepted at this time.

Thanks,\
Ahmad

---

### Note · Authors · 2023-10-19

**Comment:**

Thank you to the AE and the reviewers for your consideration of our paper.

Unfortunately we cannot agree with the comment that the paper is not well situated with respect to existing baselines as our paper primarily benchmarks against the commonly reported splits on a set of frequently cited datasets such as CommonsenseQA, StrategyQA etc. As noted in our earlier responses the Si et al paper only reports results against a randomly selected subset of samples for the datasets they report against and as such clearly isn't designed to be a "general" baseline. Moreover, as we noted, when we did investigate feasibility of adding additional comparison against Si et al we were unable to determine which specific subset they report in their paper.

In regards to the experimental setup we feel our experiments are robust and note that 3 of the 4 reviewers answered "yes" to Claims and Evidence even before we added the results of statistical significance testing and refined the wording of some of our claims as was requested.

In regards to your second bullet point, we do feel that especially after the incorporation of the reviewers good suggestions for improving clarity, the paper as it currently stands is straightforward to read.

Again, our sincere thanks to you all for the good suggestions made. We think the paper is better because of this and it is unfortunate we seem to have reached an impasse. In the interests of a quick resolution, we will withdraw our paper from TMLR at this time.

**Withdrawal Confirmation:**

I have read and agree with the venue's withdrawal policy on behalf of myself and my co-authors.